# If Virchow and Ehrlich Had Dreamt Together: What the Future Holds for *KRAS*-Mutant Lung Cancer

**DOI:** 10.3390/ijms22063025

**Published:** 2021-03-16

**Authors:** Jens Köhler, Pasi A. Jänne

**Affiliations:** 1Dana-Farber Cancer Institute, Department of Medical Oncology, Harvard Medical School, Boston, MA 02215, USA; 2Belfer Center for Applied Cancer Sciences, Boston, MA 02215, USA

**Keywords:** *KRAS*, lung cancer, NSCLC, G12C inhibitor, immunotherapy, ICI, LKB1, magic bullet

## Abstract

Non-small-cell lung cancer (NSCLC) with Kirsten rat sarcoma (*KRAS*) mutations has notoriously challenged oncologists and researchers for three notable reasons: (1) the historical assumption that *KRAS* is “undruggable”, (2) the disease heterogeneity and (3) the shaping of the tumor microenvironment by *KRAS* downstream effector functions. Better insights into *KRAS* structural biochemistry allowed researchers to develop direct *KRAS*(G12C) inhibitors, which have shown early signs of clinical activity in NSCLC patients and have recently led to an FDA breakthrough designation for AMG-510. Following the approval of immune checkpoint inhibitors for PDL1-positive NSCLC, this could fuel yet another major paradigm shift in the treatment of advanced lung cancer. Here, we review advances in our understanding of the biology of direct *KRAS* inhibition and project future opportunities and challenges of dual *KRAS* and immune checkpoint inhibition. This strategy is supported by preclinical models which show that *KRAS*(G12C) inhibitors can turn some immunologically “cold” tumors into “hot” ones and therefore could benefit patients whose tumors harbor subtype-defining STK11/LKB1 co-mutations. Forty years after the discovery of *KRAS* as a transforming oncogene, we are on the verge of approval of the first *KRAS*-targeted drug combinations, thus therapeutically unifying Paul Ehrlich’s century-old “magic bullet” vision with Rudolf Virchow’s cancer inflammation theory.

## 1. Introduction

In 1900, the German Nobel laureate Paul Ehrlich suggested a concept of “magic bullets” (“Zauberkugeln”) to specifically target invading microbes, a concept that was subsequently adapted to describe highly specific, oncogene-targeted cancer treatments [1]. More than 100 years later, non-small-cell lung cancer (NSCLC) with activating mutations of the Kirsten rat sarcoma (*KRAS*) oncogene—despite representing almost one-third of all lung cancer cases—remains a tumor entity for which no fully FDA- or EMA-approved oncogene-targeted therapies exist (for a broader overview of the frequencies of known oncogenic driver events in NSCLC we refer to [2,3,4,5,6]). Accordingly, affected patients still face a dismal prognosis [7,8,9,10]. In contrast to clinically approved oncogene-targeted therapies for various other malignancies, e.g., imatinib for BCR/ABL-positive chronic myeloid leukemia (CML) or EGFR and ALK inhibitors for EGFR-mutant and EML4/ALK-rearranged NSCLC, respectively [11,12,13,14], the development of a “magic bullet” against mutant *KRAS* has remarkably challenged scientists and physicians alike because it had long been considered “undruggable” due to biochemistry constraints [15].

Another reason for the aggressive behavior and difficulty in treating **KRAS**-mutant lung cancer is its highly inflammatory phenotype [16]. The first to postulate a connection between chronic inflammation and cancer in the 19th century was Rudolf Virchow, a pathologist at Berlin‘s Charité hospital. He had observed the presence of leucocytes (“lymphoreticular infiltrate”) in neoplastic tissues [17,18]. Unfortunately, in the century to follow, our mechanistic understanding of the bidirectional interaction between epithelial cancer and immune cells remained incomplete, and the molecular mechanisms that make inflammatory processes an important cofactor in carcinogenesis, tumor maintenance and metastasis have only recently begun to unravel [19,20].

## 2. *KRAS*-Mutant NSCLC: Therapeutically Challenging with a Plethora of Tumor Biologies

The transforming function of mutant RAS was first described by Mariano Barbacid and colleagues in 1982, and the case of a lung cancer patient with an activating *KRAS* mutation was published only two years later [21,22]. Since then, mutant *KRAS* has been identified as an important oncogenic driver for various types of solid malignancies (e.g., NSCLC, pancreatic and colorectal cancer) [23] that promotes cancer initiation, maintenance and progression in genetically engineered mouse models (GEMMs) [24,25,26]. With the general recognition of oncogene- over histology-driven tumor vulnerabilities in the early 2000s, pan-cancer sequencing efforts revealed a tissue-context-dependent distribution of mutational subtypes, with *KRAS*(G12C) being the most frequent mutation in NSCLC (45% of all *RAS* mutations), followed by *KRAS*(G12V) (~20%) and *KRAS*(G12D) (~10%) [23,27].

*KRAS* is a small GTPase that, if mutated, has a reduced ability to hydrolyze GTP or to interact with GTPase-activating proteins (GAPs). This locks *KRAS* in a GTP-bound, active state and promotes cancer cell growth and apoptosis resistance [28,29,30]. Overall, lung cancers with *KRAS* mutations are characterized by a marked disease heterogeneity: *KRAS* mutational isoforms differ in their biochemical properties to hydrolyze GTP and to activate downstream signaling pathways, which determines differences in their biological behavior and therapeutic vulnerabilities [31,32,33]. Furthermore, the presence of a wild-type *KRAS* allele affects the transforming potential of mutant *KRAS* through dimerization and impairs MEK inhibitor sensitivity [34]. Cancer cells and tumors also have variable degrees of *KRAS* dependency [35,36], and the effects of mutant *KRAS* on cellular reprogramming are tissue-context-dependent [37,38]. Finally, approximately 30% of *KRAS*-mutant tumors harbor subclass-defining co-mutations in *TP53*, *STK11*/LKB1 (the latter resulting in loss of LKB1 function) and other genes with emerging clinical and therapeutic relevance [39,40,41,42,43] (summarized in Table 1).

Etiologically, *KRAS*-mutant lung tumors are associated with a current or former smoking history (mostly *KRAS*(G12C)), but *KRAS* mutations are also found with a different mutational spectrum (mostly *KRAS*(G12D)) in up to 15% of never-smokers who develop lung cancer [44,45,46]. Most commonly, mutations are located in codon 12 (~86%), less frequently in codons 13 (~9%) and 61 (~5%) [7,47], and they occur in ~30% of adeno- and ~5% of squamous-cell carcinomas [40,48,49], even though more refined pathological analyses question the presence of *KRAS* mutations in tumors with pure squamous cell histology [50].

Despite some uncertainty regarding the prognostic impact of *KRAS* mutations due to the confounding effects of co-occurring genetic events (e.g., mutations in *STK11* or *KEAP1*), many studies suggest a more aggressive tumor biology (frequent brain/central nervous system metastases), therapeutic resistance and poorer overall survival for affected lung cancer patients compared to those with other genotypes [7,8,9,10,42,51,52,53,54].

The major caveat of pharmacologically inhibiting mutant *KRAS* had long been its high intrinsic affinity for abundant cellular GTP and the limited spatial access for small molecules to inhibit the switch-II pocket in its “OFF” state [15]. Other reasons that render *KRAS* a challenging oncogene from a therapeutic point of view are its role as a nexus of multiple downstream (MAPK, PI3K/AKT/mTOR and CDK4/6-RB) and upstream (ErbB family members, FGFR, IGFR) signaling pathways as well as the high grade of adaptational plasticity between different effector pathways [55,56,57,58,59]. Past clinical trials that have focused on targeting these effector pathways were therefore largely unsuccessful. MEK inhibitors administered on an uninterrupted schedule exhibited gastrointestinal tract- and skin-related toxicities and showed poor antitumor activity in humans despite having some activity in preclinical models [60,61]. Abemaciclib—a CDK4/6 inhibitor—also had only limited single-agent activity [62], and MEK/PI3K inhibitor combinations caused significant toxicity in humans; dose-limiting toxicities included oral mucositis, acneiform rash, hypertension, diarrhea and liver enzyme changes [63,64]. Hence, for a long time, cytotoxic chemotherapy remained the mainstay of treatment that could achieve some, but mostly short-lived, tumor control [8,54]. Therapeutic efforts have recently focused more on ERK inhibitors (e.g., GDC0994 or LY3214996) or ERK-inhibitor-based drug combinations (e.g., combined with PI3K/mTOR or CDK4/6 inhibitors), since ERK1/2 proteins are considered to have a bottleneck function in transmitting mitogenic signals and preventing MAPK pathway feedback reactivation [65,66,67,68]. These drug combinations are effective in preclinical models if applied on intermittent treatment schedules, but future clinical trials will have to clarify if this approach can overcome therapeutic limitations and toxicities observed with continuous MEK inhibition.

**Table 1 ijms-22-03025-t001:** Factors contributing to the disease heterogeneity of *KRAS*-mutant non-small-cell lung cancer (sources).

▪ KRAS mutational isoforms differ in their biochemical properties to hydrolyze GTP and to activate downstream signaling pathways. This determines their biological behavior and affects therapeutic vulnerabilities [31,32,33]. ▪ Wild-type *KRAS* protein dimerizes with mutant *KRAS*. This dimerization affects the transforming potential of mutant *KRAS* and impacts therapeutic interventions (e.g., MEK inhibition) [34]. ▪ Cancer cells and tumors have variable “RAS dependencies” [35,36]. ▪ Co-occurring genetic events like mutations in *TP53*, *STK11* and *KEAP1*, among other genes, define clinically relevant subtypes [39,40,41,42,43]. ▪ The smoking-associated etiology is the basis of a high mutational burden [69,70,71,72]. ▪ The tumor immune microenvironment ranges from T-cell-deprived (“cold”) to T-cell-inflamed (“hot”) [41,73,74,75].

## 3. Mutant *KRAS* Proteins Orchestrate the Tumor Microenvironment

The abilities of cancer cells to promote local inflammation and to simultaneously escape immune-mediated elimination are important cancer hallmarks [76]. The tumor microenvironment (TME) represents an intricate ecosystem composed of multiple noncellular and cellular components including stroma and immune cells. Cancer cells actively shape the composition and functionality of the TME by direct cell-to-cell interactions and/or by chemokine secretion. Mutant *KRAS* proteins play a central role in this process. *KRAS*-dependent effector functions increase the expression of so-called immune checkpoints like programmed death ligand-1 (PDL1), which by binding to PD1 prevents T cells from killing cancer cells [77,78]. They also restrict cancer-cell-intrinsic MHC class II expression—essential for the recognition of cancer cells by T cells [79] and impair T-cell effector functions and antitumor immunity via cyto-/chemokine-mediated (e.g., IL-1, IL-6, IL-8, GM-CSF) induction of myeloid-derived suppressor cells (MDSCs), regulatory T cells and M2-differentiated tumor-associated macrophages (TAMs) [80,81,82,83,84,85,86,87,88,89] (Figure 1). Mutant *KRAS* also induces NF-kB and cooperates with MYC—two master regulators of inflammation and immunosuppression [90,91,92,93].

Immune checkpoint inhibitors (ICIs) block the PDL1–PD1 receptor interaction and thus can reinvigorate antitumor immune responses in some patients with so-called “hot” tumors. ICIs alone or in combination with chemotherapy have become standard-of-care treatment for NSCLC patients whose tumors express PDL1 and lack *EGFR* mutations or *EML4/ALK* rearrangements [94,95,96,97,98,99]. These immunologically “hot” tumors are characterized by the accumulation of proinflammatory cytokines, high PD-L1 expression and intratumoral accumulation of CD8+ tumor-infiltrating lymphocytes (TILs), which are required for ICIs to be effective [100]. In contrast, immunologically “cold” tumors are deprived of TILs. Interestingly, *KRAS*-mutant tumors with *TP53* co-mutations are associated with a “hot” TME [73,74], whereas *STK11*/LKB1 co-mutated tumors exhibit lower expression of stimulator of interferon genes (STING) and of immune-related expression signatures than wild-type tumors (“cold” TME) and therefore are typically checkpoint-inhibitor-resistant [75,101]. Ongoing scientific efforts are seeking to decipher the mechanistic basis for this lack of ICI response in *STK11*/LKB1 co-mutated tumors with the aim of developing new strategies to turn “cold” tumors into “hot” ones and thus increase the benefit of immune checkpoint blockade for NSCLC patients with this prevalent genotype [102].

## 4. Efficacy of Direct *KRAS*(G12C) Inhibitors and Mechanisms of Resistance

The first evidence that inhibition of oncogenic *KRAS* is beneficial from a therapeutic viewpoint came from immunocompetent genetically engineered mouse models (GEMMs) of lung cancer in which lung tumors completely regressed upon genetic removal of mutant *KRAS* [24,25,26]. In the last decade, better biochemical insights into *KRAS* structural biology finally allowed researchers to develop direct *KRAS*(G12C) inhibitors like the preclinical compounds SML-8-73-1, compound 12 (“Shokat compound”), ARS-853 and ARS-1620, as well as the clinical compounds AMG-510 (Amgen), MRTX-849 (Mirati Therapeutics) and JNJ-74699157/ARS-3248 (Johnson & Johnson/Wellspring Biosciences) [103,104,105,106,107,108,109,110]. These inhibitors bind covalently to the cysteine residue in the switch-II pocket, which is newly created by the G12C mutation. Binding of the inhibitor shifts the relative nucleotide affinities to favor GDP over GTP binding and reduces the interactions between mutant *KRAS* and effector or regulatory molecules [107].

The phase I/II CodeBreaK-100 trial (NCT03600883) investigating the clinically most advanced *KRAS*(G12C) inhibitor AMG-510 (international nonproprietary name (INN) sotorasib) reported a confirmed objective response rate (ORR) of ~32% and a disease control rate (DCR) of ~88% among lung cancer patients for the phase I trial part. Grade 3/4 toxicities and treatment discontinuation occurred in 11.6% and 7% of patients, respectively [111]. Based on this study, in December 2020, the FDA granted breakthrough therapy designation for sotorasib for patients with *KRAS*(G12C)-mutant, locally advanced or metastatic NSCLC following at least one prior systemic therapy. The phase II trial part with a data cutoff on 1 December 2020 and a median follow-up of 12.2 months validated the phase I results with a confirmed ORR of 37.1%, a DCR of 80.6% and a median duration of response of 10 months. The median progression-free survival was 6.8 months. Treatment-related adverse events (TRAEs) led to treatment discontinuation in 7.1% of patients. Most TRAEs were grade 1 and 2 and included diarrhea (31% any grade), nausea (19%), liver enzyme changes (15%) and fatigue (11%) (presented at the 2020 World Conference on Lung Cancer, 28–31 January 2021).

The median follow-up of the phase I/II KRYSTAL-1 trial (NCT03785249) for Mirati’s MRTX-849 compound (INN adagrasib) was still relatively short (9.6 months) at the last study update presented at the 32nd EORTC-NCI-AACR Symposium (24–25 October 2020), but the data reported for NSCLC patients appear slightly better than those for AMG-510, although longer follow-up is still required [112]. Of the evaluable patients, 45% achieved a partial response, and the DCR was 96%. TRAEs led to treatment discontinuation in 4.5% of patients and most commonly included nausea (54%), diarrhea (48%), vomiting (34%), fatigue (28%) and increased liver enzymes (23%). The only commonly reported grade 3/4 TRAE was hyponatremia (3%) (the clinical efficacy parameters are summarized in Table 2). Adagrasib has a longer half-life of about 24 h compared to sotorasib (half-life 6.5 h), which allows for continuous drug exposure and sustained *KRAS* target inhibition. Adagrasib also penetrates the blood–brain barrier in murine models and showed efficacy in one patient with NSCLC and active brain metastases, an important feature for this highly metastatic disease with frequent brain/central nervous system manifestation.

It is most likely, however, that allele-specific *KRAS*(G12C) inhibitors will be combined with other treatment modalities. Monotherapies presumably have limited long-term efficacy in terms of preventing adaptive resistance since (1) tumors with minor fractions of cancer cells that harbor non-G12C *KRAS* mutations will ultimately relapse due to selection of these subclones [113,114] and (2) several mechanisms have been proposed for how cancer cells can lose their *KRAS* dependency. Among others, these include YAP pathway activation [115] and increased *KRAS*(G12C) expression via EGFR or aurora kinase signaling [116]. Drug combinations can furthermore account for the fact that currently available *KRAS*(G12C) inhibitors only target the inactive, GDP-bound form of *KRAS* and thus rely on the residual intrinsic hydrolysis of GTP to revert *KRAS* into the GDP-bound state. This mechanism is vulnerable to adaptive responses that activate upstream signaling, e.g., via receptor tyrosine kinases (RTKs) like the ErbB family or FGFR [56,58,117]. These RTKs signal through SHP2 (encoded by *PTPN11*), increase the GTP-loaded “ON” form of *KRAS* and therefore reduce *KRAS*(G12C) inhibitor target engagement [118,119]. Inhibition of SHP2 reduces this conversion of GDP- into GTP-bound *KRAS* and overcomes adaptive resistance to MAPK-pathway-targeted agents including *KRAS*(G12C) inhibitors [120,121]. In a patient with NSCLC, a reduction of tumor volume has been observed when adagrasib was combined with the experimental SHP2 inhibitor TNO-155 (Novartis), and the corresponding phase I/II KRYSTAL-2 study is currently recruiting patients (NCT04330664). In another phase Ib clinical trial, the combination of sotorasib and the experimental SHP2 inhibitor RMC-4630 will be investigated [122,123]. Other drug combinations (adagrasib plus pan-ErbB inhibitor afatinib; ARS-1620 plus mTOR inhibitor (everolimus) or linsitinib) follow the same biological rationale of simultaneously inhibiting *KRAS* and the nucleotide exchange on *KRAS* via RTKs [124] (for a comprehensive overview of clinical trials investigating strategies to overcome resistance to drugs targeting the *KRAS*(G12C) mutation we refer to [125]).

## 5. Conclusions and Future Perspectives of Combination Therapeutic Approaches

More than a century after the pioneering scientific work of Rudolf Virchow and Paul Ehrlich [1,17,18], for non-small-cell lung cancer (NSCLC), the era of cancer immunotherapy—which harnesses the immune system to kill cancer cells—follows the era of groundbreaking discoveries in the field of oncogene-targeted therapies [19,20]. However, the progress made during the “targeted therapy revolution” for *EGFR*-mutant and *EML4/ALK*-rearranged lung cancers among other oncogene-addicted pulmonary malignancies (for a comprehensive overview of oncogene-directed therapies against NSCLC, e.g., with ROS1/NTRK, BRAF, MET and HER-2 aberrations, we refer to [126]) had largely spared *KRAS*-mutant NSCLC despite the anticipated efficacy of *KRAS* inhibitors for this highly prevalent but heterogeneous lung cancer subtype [11,12,14,24,25,26,127] (Table 1). After overcoming biochemistry constraints to directly inhibit *KRAS*(G12C), the most frequent mutational subtype in NSCLC, the historical assumption of *KRAS* as being an “undruggable” target needs to be irrevocably discarded [103,104,105,106,107,108,109,110]. First reports from phase I/II clinical trials investigating the direct *KRAS*(G12C) inhibitors sotorasib and adagrasib are very impressive considering this notoriously hard-to-treat patient subgroup. Response rates are slightly inferior to those observed with other oncogene-targeted therapies (e.g., against mutant EGFR or rearranged EML4/ALK) in pretreated patients, and differences in response rates could be due to the heterogeneity of *KRAS*-mutant lung tumors with multiple DNA-damage-associated genomic alterations [128,129,130,131]. Additional *KRAS*(G12C) inhibitors are continuing to emerge for which clinical efficacy parameters have yet to be reported (JNJ-74699157/ARS-3248: NCT04006301; GDC-6036: NCT04449874); for others, the clinical development has been stopped due to unexpected toxicities (LY-3499446: NCT04165031). Currently ongoing (CodeBreaK-200 for AMG-510: NCT04303780) and future randomized phase III trials will ultimately show the true benefit of direct *KRAS*(G12C) inhibitors in untreated patients and presumably establish *KRAS*(G12C) inhibitors as the frontline treatment for *KRAS*(G12C)-mutant lung cancers (the current status of clinical development of *KRAS*(G12C) inhibitors is summarized in Table 2).

To induce deeper initial tumor regressions and to prevent the emergence of resistant cancer cell clones, multidrug combinations (e.g., *KRAS*(G12C) inhibitors with SHP2 and pan-ErbB inhibitors) are currently being clinically evaluated. Historically, drug combinations targeting *KRAS*-dependent downstream pathways (e.g., continuous MEK and PI3K inhibition) have been limited by toxicities [63,64], but *KRAS*(G12C) inhibitors avoid wild-type *KRAS* reactivity and therefore are less prone to off-target effects that are believed to disturb tissue homeostasis [127]. The lack of dose-limiting toxicities observed with sotorasib and adagrasib is encouraging and seems to make them ideal partners for combination treatment strategies, including those that incorporate immunotherapy.

Sensitivity to immune checkpoint inhibitors (ICIs) has been associated with a high tumor mutational burden (TMB) [132,133,134], and therefore, the smoking-related etiology of *KRAS*(G12C)-mutant NSCLC with a high mutational burden predestines affected patients for immunotherapy [44,45,46,69,70,71,72,135]. ICIs have been established as standard-of-care treatment for NSCLC patients whose tumors express PDL1 and lack *EGFR* mutations or *EML4/ALK* rearrangements as a single agent or in combination with chemotherapy [94,95,96,97,134]. However, response rates to single-agent ICIs overall are modest, and strategies to overcome this limitation are urgently required. The clinical benefit of combined PD1/PDL1 and CTLA-4 inhibition remains controversial despite FDA approval of the ipilimumab plus pembrolizumab combination [136] and comes at the cost of an increased risk of serious immune-related adverse events compared to anti-PD-1 therapy alone [137].

Due to the important function of *KRAS* in reducing cancer cell immunogenicity and inducing local immunosuppression (Figure 1), *KRAS*(G12C) inhibitors were expected to have profound effects on the tumor microenvironment (TME). In *KRAS*-mutant NSCLC, the TME is frequently characterized by a paucity, lack and/or dysfunction of tumor-infiltrating leukocytes (TILs), especially in the presence of co-occurring mutations in *STK11*/LKB1 [41,75,101]. This immunologically “cold” TME impairs the efficacy of ICIs [100], and therefore, strategies to turn “cold” tumors into “hot” ones are urgently needed. Indeed, similarly to MEK and SHP2 inhibition, sotorasib and adagrasib induced a more proinflammatory and TIL-infiltrated TME in mouse models (“reconditioning” effect) [103,121,138,139,140]. This translated into durable complete responses in combination with anti-PD-1 therapy. Mice that were cured with a combination of sotorasib and pembrolizumab subsequently rejected *KRAS*(G12C)-mutant CT26 tumors, suggesting that combining *KRAS*(G12C) inhibitors with immune checkpoint inhibitors could even drive an acquired immune response. Adaptive rather than innate immunity offers the greatest potential for durable, robust anticancer immune responses to prevent a tumor relapse and/or metastatic spread. However, these results from preclinical models have yet to be confirmed in human clinical trials. The combination of a *KRAS*(G12C) inhibitor and immunotherapy could specifically benefit those patients whose tumors harbor *STK11/LKB1* co-mutations (~30% of *KRAS*(G12C)-mutant NSCLC). These tumors are linked to poor outcomes with immunotherapy and platinum-based chemotherapy [75,141,142]. Even though numerous strategies aimed at bolstering immunity against *STK11*-mutant tumors are currently under investigation (e.g., dual immune checkpoint inhibition with nivolumab and ipilimumab [143]), a broader spectrum of efficacious therapeutic options is urgently needed. Exploratory correlative analyses from the KRYSTAL-1 (presented at the 32nd EORTC-NCI-AACR Symposium, 24–25 October 2020) and CodeBreaK 100 (presented at the 2020 World Conference on Lung Cancer, 28–31 January 2021) trials in this context suggest higher response rates for single-agent adagrasib (64% versus 45%) and sotorasib (50% versus 42%) among patients whose tumors also harbored an *STK11*/LKB1 co-mutation. Even though these early findings need to be confirmed in larger clinical trials that combine sotorasib (NCT03600883) and adagrasib (NCT04613596, KRYSTAL-7) with the anti-PD-1 antibody pembrolizumab, they give us a glimpse of the extraordinary potential of these *KRAS*-targeted agents for this historically difficult-to-treat patient subgroup.

Other strategies to boost the immune response against *KRAS*-mutant cancers include STING agonists (ADU-S100, MK-1454) [101,144,145], as well as CAR-T cells (adoptive T-cell transfer) [146] and mRNA vaccine technology [147]. For the latter, the current worldwide first use of mRNA vaccine technology to fight the COVID-19 pandemic [148,149] could boost its development and acceptance in the field of oncology. In an ongoing phase I trial (V941-001), Moderna and Merck are testing mRNA-5671 alone and in combination with pembrolizumab in patients with *KRAS*-mutant cancers. mRNA-5671 is designed to generate and present the four most prevalent *KRAS* mutations (G12C, G12D, G12V and G13C) as neoantigens in host cells to the immune system to drive a more robust T-cell response (no efficacy data are publicly available yet).

A major caveat we still face today is the fact that so far no specific inhibitors of non-G12C mutations have entered clinical trials. In NSCLC, these mutations represent more than 50% of all *KRAS* mutations. A potential first-in-class inhibitor of *KRAS*(G12D), MRTX-1133, is currently in preclinical development by Mirati (to the best of our knowledge, there are no publicly available data on this compound yet). In an alternative approach, the son of sevenless 1 (SOS1) protein, which determines the nucleotide exchange on *KRAS*, has gained much attention as a therapeutic target to inhibit all major G12D/V/C and G13D variants. Boehringer Ingelheim’s BI-1701963 and Bayer’s BAY-293 “pan-KRAS inhibitors” selectively inhibit the SOS1–KRAS interaction [150,151], but unfortunately, apoptosis induction and tumor regressions were only observed when this drug class was combined with a MEK inhibitor. Phase I dose-finding studies are currently recruiting patients with solid tumors (BI-1701963 plus trametinib: NCT04111458) or are planned (BI-1701963 plus adagrasib). Other strategies to target non-G12C mutations include so-called switch I/II pocket inhibitors like BI-2852, which bind with nanomolar affinities to the active and inactive forms of *KRAS* [152,153], or mutant-selective “tricomplex” inhibitors, which sterically block interactions between *KRAS* and effector proteins such as RAF [154]. *KRAS*-targeting monobodies [155] and intrabodies [156] further add to the spectrum of therapeutic approaches currently under investigation.

From an oncologist’s perspective, the coming years in the field of *KRAS*-mutant NSCLC will be exciting and extremely laborious at the same time. Future clinical trials will have to teach us which drug combinations out of the multiple available therapeutic concepts (summarized in Figure 2) are the most efficacious ones and/or which treatment sequence is optimal from a tumor evolution perspective. The establishment of potent treatment predictors and systematic analysis of on-treatment longitudinal biopsies will hold the key to a better understanding of the biology of treatment resistance and guide clinical decision-making about rationally designed subsequent treatment combinations. To speed up the efficient development of drug combinations, the refinement of *KRAS*-mutant GEMMs to better recapitulate the genetic and immunologic complexity of human lung tumors with a high tumor mutational burden is absolutely desirable [157,158].

Despite some limitations, it is extremely exciting to see that more than a century after the groundbreaking scientific work of Paul Ehrlich and Rudolf Virchow, we are now on the verge of therapeutically unifying the concepts of both pioneers to harness synergistic effects between immune checkpoint inhibitors with “magic bullet” *KRAS*(G12C) inhibitors that have the potential to “recondition” the immunosuppressed tumor microenvironment. More than 40 years after the identification of RAS as a transforming oncogene, this approach will revolutionize paradigms for *KRAS*-mutant NSCLC. Therefore, after many discouraging therapeutic attempts in the past, we have every reason to look to the future with optimism.

## Figures and Tables

**Figure 1 ijms-22-03025-f001:**
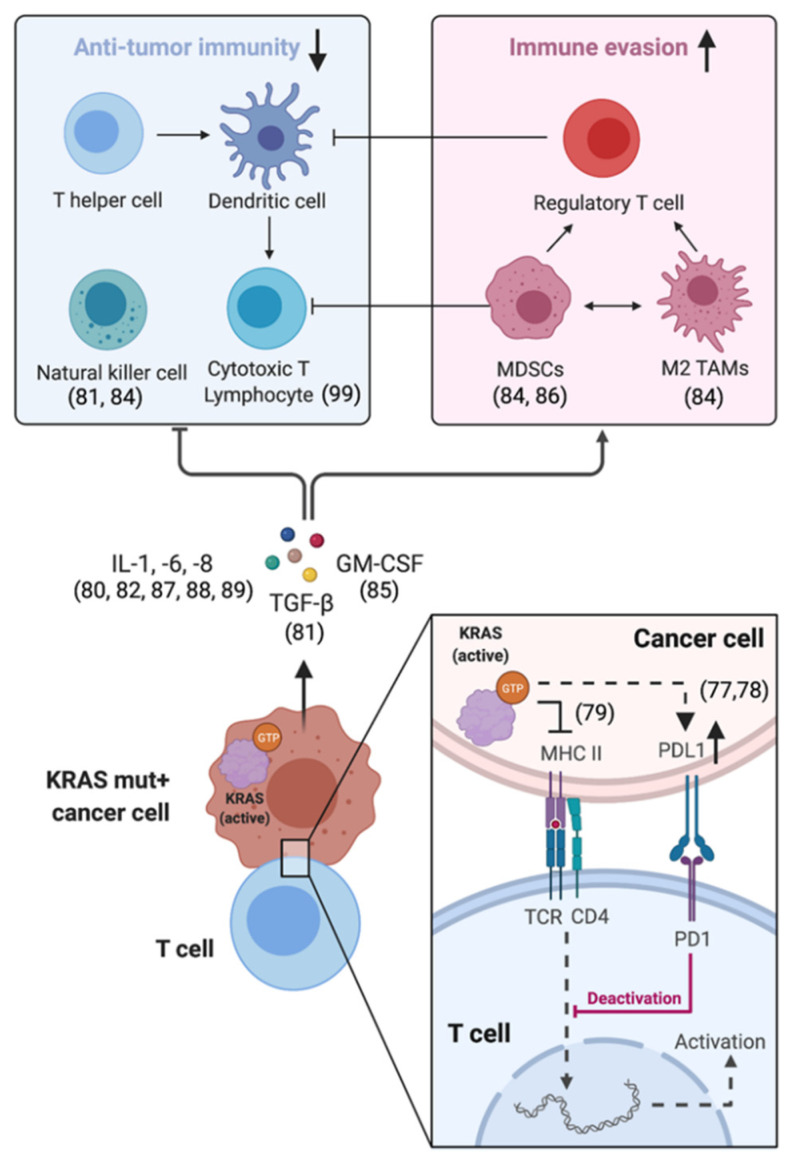
Simplified overview of mutant-KRAS-dependent effects on the surrounding tumor microenvironment via direct cell-to-cell interactions and/or paracrine secretion of interleukins, GM-CSF and TGFβ. These paracrine signals induce the accumulation of myeloid-derived suppressor cells (MDSCs), M2-differentiated tumor-associated macrophages (TAMs) and regulatory T cells, which impair antitumor immunity by suppressing T-cell effector functions. References are displayed in brackets.

**Figure 2 ijms-22-03025-f002:**
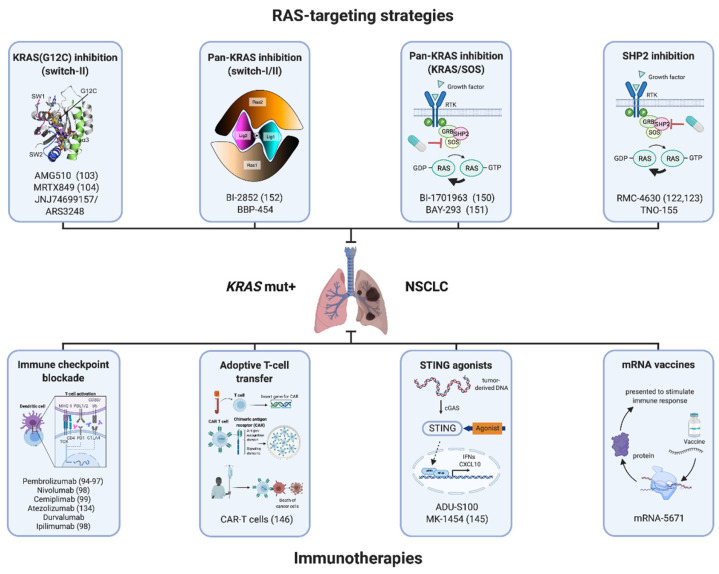
Overview of therapeutic concepts for patients with *KRAS*-mutant NSCLC in different stages of clinical development that target mutant *KRAS* itself or the surrounding tumor immune microenvironment. Future clinical trials are required to decipher the optimal strategy of simultaneously or sequentially combining these treatment strategies. References if applicable are displayed in brackets. Created with biorender.com (accessed on 3 March 2021).

**Table 2 ijms-22-03025-t002:** Summary of ongoing clinical trials investigating *KRAS*(G12C) inhibitors alone or in combination with other treatment modalities in NSCLC. ORR = objective response rate, PFS = progression-free survival, TRAE = treatment-related adverse events.

Drug	Trial #	Clinical Phase	Efficacy (%)	Median PFS	Reported Toxicity Any Grade in % (Grade 3–4 in %)	Data Cutoff
Sotorasib (AMG510) −/+pembrolizumab	NCT03600883 CodeBreak 100	Phase 1/2 recruiting	ORR 37.1 Complete response 2.4 Partial response 34.7 Stable disease 43.5Progressive disease 16.1 Disease control rate 80.6	6.8 months	Diarrhea 69.8 (19.8) Nausea 19.0 (0) ALT increase 15.1 (6.3) AST increase 15.1 (5.6) Fatigue 11.1 (0) Vomiting 7.9 (0) Rash 5.6 (0) Treatment discontinuation in 7.1%	1 December 2020 with a median follow-up time of 12.2 months
Sotorasib (AMG510) +MEK inhibitor or anti-PD1	NCT04185883 CodeBreak 101	Phase 1b recruiting	-	-	-	-
Sotorasib (AMG510) in subjects of Chinese descent	NCT04380753 CodeBreak 105	Phase 1 recruiting	-	-	-	-
Sotorasib (AMG510) vs. docetaxel	NCT04303780 CodeBreak 200	Phase 3 recruiting	-	-	-	-
Adagrasib (MRTX849) −/+pembrolizumab −/+afatinib	NCT03785249 KRYSTAL-1	Phase 1/2 recruiting	ORR 45 Complete response 0 Partial response 45 Stable disease 51 Progressive disease 2 Disease control rate 96	-	Nausea 54 (2) Diarrhea 51 (2) Vomiting 35 (2) Fatigue 32 (6) Increased ALT 20 (5) Increased AST 17 (5) Increased creatinine 15 (0) Decreased appetite 15 (0) QT prolongation 14 (3) Anemia 13 (2) Grade 5 TRAEs in two patients (pneumonitis, cardiac failure) Discontinuation due to TRAEs in 4.5%	30 August 2020 with a median follow-up time of 9.6 months
Adagrasib (MRTX849) +TNO155	NCT04330664 KRYSTAL-2	Phase 1/2 recruiting	-	-	-	-
Adagrasib (MRTX849) +pembrolizumab	NCT04613596 KRYSTAL-7	Phase 2 recruiting	-	-	-	-
GDC-6036 −/+atezolizumab −/+bevacizumab −/+erlotinib	NCT04449874	Phase 1a/1b recruiting	-	-	-	-
LY3499446 +/− abemaciclib/ erlotinib vs. docetaxel	NCT04165031	Phase 1 terminated due to toxicity	-	-	-	-

## Data Availability

Not applicable.

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
