# Peer review of "If Virchow and Ehrlich Had Dreamt Together: What the Future Holds for KRAS-Mutant Lung Cancer"

_ijms, 2021, doi:10.3390/ijms22063025_

Round 1

Reviewer 1 Report

The manuscript by Kohler and co-author addresses the role of KRAS oncogene mutations in non small cell lung cancer, and the potential therapeutic role of KRAS inhibitors either alone and in combination with other treatment strategies, including immunotherapy.

The following are my comments:

  • Section 2, lines 71-82: The Authors should specify that KRAS oncogene has a relevant role in several types of solid tumors, besides NSCLC. Please also consider to provide a wider overview of molecular alterations in lung cancer, specifying the frequency of each druggable mutation along with KRAS mutations. Also, in this section the authors should mention clinical features of KRAS mutant disease (e.g. frequent brain/CNS metastasis), and the pattern of response observed during standard treatment(s), which are partly reported in section 4.

  • In Section 2, the Authors should consider to provide explanations on what makes KRAS such a unique and “challenging” oncogene to target, including alternative oncogenic pathways for resistance.

  • Section 2, line 96: the Authors mention the problem of toxicities related to continuous MEK inhibition as a potential limiting factor for the development of these drugs’ combination. Please provide more details on this treatment related toxicity, as it has represented a relevant obstacle in the development of targeted drugs inhibiting this plethoric signalling pathway.

  • Section 3, lines 101-102: the Authors correctly cite the immune system as an important element among the hallmarks of cancer (Hanahan et al., Cell 2011). Yet, before explaining the role of KRAS in promoting a pro-inflammatory state, the Authors should better explain the equilibrium among immune cells and cancer cells, including the difference between “hot” and “cold” tumors. Moreover, it should be better explained how this can be exploited by drugs acting as immune checkpoint inhibitors, which are already standard of care, and how their anti-tumor efficacy can be increased in KRAS mutant tumors.

  • Section 4, line 129: remove the comma sign after “First evidence” at the beginning of the sentence.

  • Section 4, lines 141-163: the Authors present important preliminary data on drugs targeting KRAS. Please provide more details on the safety profile, including the reason(s) for treatment discontinuation, the rate and characteristics of serious adverse events. The issue of toxicity is relevant, as it had represented a challenge for the development of KRAS inhibitors for several years.

  • Section 4: A table reporting the features of different KRAS inhibitors might allow a comparison regarding pharmacologic characteristics, ORR, DCR, median follow up of available studies, and incidence of adverse events.

  • Section 4, lines 164-180: this last part of the paragraph is confusing and should be modified. What do the Authors mean with the sentences in lines 164-171? Are they talking about KRAS mutant tumors or about all-comers NSCLC treated with immunotherapy? Has this role of KRAS inhibitors been demonstrated in tumor cell or animal models, and has it been confirmed in combination studies with immunotherapy? Overall, I would suggest to move the explanation on the role of KRAS inhibitors as immunomodulating agents in the previous sections. Here, the Authors should only discuss preclinical and, if any, clinical data of efficacy of the combination of KRAS inhibitors and immune-checkpoint inhibitors.

  • The whole Section 5 should be modified, moving all references of molecular mechanisms of treatment resistance and/or intrinsic pharmacologic mechanisms of KRAS inhibitors in a dedicated previous section. Here in the conclusive part, I would suggest to discuss and present the current therapeutic strategies, the ongoing research and all the future perspectives, including combination therapies.

  • Section 5, lines 184-185: the Authors mention EGFR and ALK inhibitors, but also ROS, BRAF, RET, MET, HER-2, NTRK targeted agents should be mentioned as important targeted therapies which have been recently introduced and significantly modified the treatment strategy.

  • Section 5, lines 187-190: Please consider moving this sentence in previous sections regarding the molecular characteristics of KRAS oncogenic pathway and the reasons why it is a challenging pharmacological target.

  • Section 5, line 196: please find a different definition for response rates of KRAS inhibitors, as “shy” does not sound as a proper scientific definition.

  • Section 5, lines 208-215: I agree to discuss here the potential role of combination treatments, however the Authors should consider to move the discussion on molecular mechanisms for adaptive resistance in the previous part of the manuscript, where it was already discussed.

  • Section 5, line 225: what do the Authors mean with “NSCLC in general and KRAS mutant NSCLC are predestined for immunotherapy”. This is not clear, please explain.

  • This final section is very interesting as it provides details on ongoing clinical research in the field of KRAS mutant therapy and combination strategies. A table reporting main clinical trials currently underway might make this part more clear and provide more precise data. Also, the Authors should provide a critical and personal view on the current available evidences in this field and on ongoing research, to add more value to the manuscript.

Author Response

Answers to Reviewer 1 regarding manuscript IJMS-1112527

We sincerely thank Reviewer 1 for her/his considerate comments, scientific input and editorial suggestions regarding our review manuscript.

Please find below our detailed answers to each point raised by Reviewer 1:

Reviewer 1

The manuscript by Kohler and co-author addresses the role of KRAS oncogene mutations in non small cell lung cancer, and the potential therapeutic role of KRAS inhibitors either alone and in combination with other treatment strategies, including immunotherapy.

The following are my comments:

  • Section 2, lines 71-82: The Authors should specify that KRAS oncogene has a relevant role in several types of solid tumors, besides NSCLC. Please also consider to provide a wider overview of molecular alterations in lung cancer, specifying the frequency of each druggable mutation along with KRAS mutations. Also, in this section the authors should mention clinical features of KRAS mutant disease (e.g. frequent brain/CNS metastasis), and the pattern of response observed during standard treatment(s), which are partly reported in section 4.

Answer: In the revised manuscript version, we mention the important role of mutant KRAS as oncogenic driver also for other tumor entities (e.g. pancreatic and colorectal cancer) and additionally cite reference 23 (section 2, lines 55-57 of the untracked manuscript version). Regarding the suggestion of reviewer 1 to include a wider overview on oncogenic driver events in NSCLC, we refer to several publications addressing this topic (section 1, lines 35-36 of the untracked manuscript version, references 2-6). Our intention for this review article was not to provide a general overview on the genomic landscape of NSCLC but rather to focus on NSCLC with KRAS mutations.

As suggested, we furthermore added the aggressive tumor biology with frequent brain/CNS metastases and therapeutic resistance to the manuscript (section 2, lines 86-88 of the untracked manuscript version). However, it is noteworthy, that in contrast to colorectal cancer, for which an increased risk for metastatic spread in the presence of KRAS mutations has been clearly reported (Tie et al., Clinical Cancer Research, 2011, DOI: 10.1158/1078-0432.CCR-10-1720. “KRAS Mutation Is Associated with Lung Metastasis in Patients with Curatively Resected Colorectal Cancer.”), this effect on metastatic risk in general and to the CNS/brain in particular remains more controversial for non-small cell lung cancer and is potentially confounded by mutations in STK11or KEAP1 (El Costa et al., J Thorac Oncology, 2019, DOI: 10.1016/j.jtho.2019.01.020. “Characteristics and Outcomes of Patients With Metastatic KRAS-Mutant Lung Adenocarcinomas: The Lung Cancer Mutation Consortium Experience.“; Arbour et al., Clin Cancer Res, 2018, DOI: 10.1158/1078-0432.CCR-17-1841. “Effects of Co-occurring Genomic Alterations on Outcomes in Patients with KRAS-Mutant Non-Small Cell Lung Cancer.”; Offin et al., Cancer, 2019, doi: 10.1002/cncr.32461. “Frequency and outcomes of brain metastases in patients with HER2‐mutant lung cancers”; Zhao et al., Lung Cancer, 2014, DOI: 10.1016/j.lungcan.2014.08.013 “Alterations of LKB1 and KRAS and risk of brain metastasis: comprehensive characterization by mutation analysis, copy number, and gene expression in non-small-cell lung carcinoma.”).

  • In Section 2, the Authors should consider to provide explanations on what makes KRAS such a unique and “challenging” oncogene to target, including alternative oncogenic pathways for resistance.

Answer: Regarding the “challenge” that mutant KRAS specifically poses as a therapeutic target, in our initial manuscript submission, we had already mentioned biochemistry constraints of developing direct KRAS inhibitors: 1) the high affinity of mutant KRAS for abundant cellular GTP and the lack of surfaces to which an inhibitor can bind. In the revised version, we also emphasize the role of KRAS as a nexus of multiple downstream (MAPK, PI3K/AKT/mTOR and CDK4/6-RB) and upstream (ErbB family members, FGFR, IGFR) signaling pathways as well as the high grade of adaptational plasticity between different effector pathways (section 2, lines 90-96 of the untracked manuscript version).

  • Section 2, line 96: the Authors mention the problem of toxicities related to continuous MEK inhibition as a potential limiting factor for the development of these drugs’ combination. Please provide more details on this treatment related toxicity, as it has represented a relevant obstacle in the development of targeted drugs inhibiting this plethoric signalling pathway.

Answer: In the revised manuscript, we describe more in detail the side effects that were observed during clinical trials with continuous MEK inhibition (most frequently gastrointestinal tract and skin related toxicities, references 60 and 61) as well as oral mucositis, liver enzyme elevation, acneiform rash, hypertension, and diarrhea as the most frequent dose-limiting toxicities observed with combined MEK plus PI3K inhibition (references 63 and 64).

  • Section 3, lines 101-102: the Authors correctly cite the immune system as an important element among the hallmarks of cancer (Hanahan et al., Cell 2011). Yet, before explaining the role of KRAS in promoting a pro-inflammatory state, the Authors should better explain the equilibrium among immune cells and cancer cells, including the difference between “hot” and “cold” tumors. Moreover, it should be better explained how this can be exploited by drugs acting as immune checkpoint inhibitors, which are already standard of care, and how their anti-tumor efficacy can be increased in KRAS mutant tumors.

Answer: We have substantially changed this paragraph and explain now more in detail the mechanistic basis of the concept of immune checkpoint inhibition and the differentiation of immunologically “cold” (T-cell deprived tumors) versus “hot” (T-cell enriched tumors with expression of pro-inflammatory cytokines and expression of immune-related gene signatures) tumor microenvironments. We also cite major publications that led to the approval of immune checkpoint inhibitors alone or in combination with chemotherapy (section 3, references 94-98).

  • Section 4, line 129: remove the comma sign after “First evidence” at the beginning of the sentence.
  • Answer: We have removed the comma from this sentence.

  • Section 4, lines 141-163: the Authors present important preliminary data on drugs targeting KRAS. Please provide more details on the safety profile, including the reason(s) for treatment discontinuation, the rate and characteristics of serious adverse events. The issue of toxicity is relevant, as it had represented a challenge for the development of KRAS inhibitors for several years.

Answer: We thank Reviewer 1 for this comment since we also believe that the toxicity profile is of utmost importance not only for single agent but especially for combinatorial treatment approaches. Therefore, in the revised manuscript, we describe more in detail the observed side effects for both clinically evaluated KRAS(G12C) inhibitors (section 4, AMG510/sotorasib: lines 176-179, MRTX849/adagrasib: lines 185-188).

  • Section 4: A table reporting the features of different KRAS inhibitors might allow a comparison regarding pharmacologic characteristics, ORR, DCR, median follow up of available studies, and incidence of adverse events.

Answer: We added a Table 1 which summarizes the stage of clinical development and (if available) the reported clinical efficacy and toxicity characteristics of direct KRAS(G12C) inhibitors as single agent or in combination with other treatment modalities.

  • Section 4, lines 164-180: this last part of the paragraph is confusing and should be modified. What do the Authors mean with the sentences in lines 164-171? Are they talking about KRAS mutant tumors or about all-comers NSCLC treated with immunotherapy? Has this role of KRAS inhibitors been demonstrated in tumor cell or animal models, and has it been confirmed in combination studies with immunotherapy? Overall, I would suggest to move the explanation on the role of KRAS inhibitors as immunomodulating agents in the previous sections. Here, the Authors should only discuss preclinical and, if any, clinical data of efficacy of the combination of KRAS inhibitors and immune-checkpoint inhibitors.

Answer: The sentences in lines 164-171 of the first manuscript submission elaborate on the clinical efficacy parameters of direct KRAS(G12C) inhibitors which have only been tested in patients with KRAS(G12C) mutations. This part does not relate to combination treatments with immunotherapies. We have also renamed section 4 to “Efficacy of direct KRAS(G12C) inhibitors and mechanisms of resistance” in which we exclusively talk about the pre-clinical and clinical evidence of the efficacy of direct KRAS(G12C) inhibitors and potential mechanisms of resistance which have so far mostly been investigated in pre-clinical tumor models. We also moved the concept of simultaneous inhibition of mutant KRAS itself and of the nucleotide exchange on KRAS via RTKs and SHP2 from the Conclusions part into this manuscript part since it refers to mechanisms which impair single agent efficacy of direct KRAS(G12C) inhibitors.

  • The whole Section 5 should be modified, moving all references of molecular mechanisms of treatment resistance and/or intrinsic pharmacologic mechanisms of KRAS inhibitors in a dedicated previous section. Here in the conclusive part, I would suggest to discuss and present the current therapeutic strategies, the ongoing research and all the future perspectives, including combination therapies.

Answer: As suggested, we have moved major parts including mechanisms of resistance to direct KRAS inhibitors from section 5 to section 4.

  • Section 5, lines 184-185: the Authors mention EGFR and ALK inhibitors, but also ROS, BRAF, RET, MET, HER-2, NTRK targeted agents should be mentioned as important targeted therapies which have been recently introduced and significantly modified the treatment

Answer: We thank Reviewer 1 for her/his suggestion. We added other oncogene-addicted lung cancers (e.g. with ROS, BRAF, RET, MET, HER-2, NTRK aberrations) to the “Conclusions and Future Perspectives” part of the manuscript and refer to a comprehensive review article on oncogene directed therapies in NSCLC by Yuan et al. “The emerging treatment landscape of targeted therapy in non-small-cell lung cancer.” Signal Transduct Target Ther 4, 61 (2019). Since the scope of our review was to focus on KRAS targeted therapies in particular and not oncogene-directed therapies in general, EGFR and ALK inhibitors are exemplarily mentioned as inhibitors for frequently occurring driver events in non-small cell lung cancer. 

  • Section 5, lines 187-190: Please consider moving this sentence in previous sections regarding the molecular characteristics of KRAS oncogenic pathway and the reasons why it is a challenging pharmacological target.

Answer: We moved this sentence to Section 2.

  • Section 5, line 196: please find a different definition for response rates of KRAS inhibitors, as “shy” does not sound as a proper scientific definition.

Answer: We have replaced the words “shy of” with “slightly inferior to”.

  • Section 5, lines 208-215: I agree to discuss here the potential role of combination treatments, however the Authors should consider to move the discussion on molecular mechanisms for adaptive resistance in the previous part of the manuscript, where it was already discussed.

Answer: As suggested, we have moved major parts including mechanisms of resistance to direct KRAS inhibitors from section 5 to section 4.

  • Section 5, line 225: what do the Authors mean with “NSCLC in general and KRAS mutant NSCLC are predestined for immunotherapy”. This is not clear, please explain.

Answer: In the revised version of the manuscript, we changed the structure of the sentence to clarify that immunotherapy efficacy has been associated with an increased tumor mutational burden (TMB) which - due to the smoking-related etiology of KRAS(G12C) mutant lung cancers, renders affected patients a preferable target population to receive immune checkpoint inhibitors.   

  • This final section is very interesting as it provides details on ongoing clinical research in the field of KRAS mutant therapy and combination strategies. A table reporting main clinical trials currently underway might make this part more clear and provide more precise data. Also, the Authors should provide a critical and personal view on the current available evidences in this field and on ongoing research, to add more value to the manuscript.

Answer: We added a Table 1 which summarizes the stage of clinical development and (if available) the reported clinical efficacy and toxicity characteristics of direct KRAS(G12C) inhibitors as single agent or in combination with other treatment modalities.

Reviewer 2 Report

This is a very well written review, with strong and up-to-date scientific content. Importantly, it is also interesting and intellectually challenging outside its molecular aspects. I very much liked the historical context brought in by authors.

I believe, the paper is ready for publication. I have few minor remarks for your consideration.

Page 2 vv 63-68  – the differences between KRAS mutational isoforms are implied, but no specific details are mentioned, therefore it is not clear what is their biological importance.

This is true for Fig1 as well . It is not clear what properties exactly are implied or how wtKRAS affects mKRAS biology etc etc

Fig 2 and 3  the description is very vague, references in the text not well marked either. As a result both figures are not as helpful for the reader.

Author Response

Answers to Reviewer 2 regarding manuscript IJMS-1112527

We sincerely thank Reviewer 2 for her/his considerate comments, scientific input and editorial suggestions regarding our review manuscript.

Please find below our detailed answers to each point raised by Reviewer 2:

Reviewer 2

This is a very well written review, with strong and up-to-date scientific content. Importantly, it is also interesting and intellectually challenging outside its molecular aspects. I very much liked the historical context brought in by authors. I believe, the paper is ready for publication. I have few minor remarks for your consideration.

Page 2 vv 63 - 68  – the differences between KRAS mutational isoforms are implied, but no specific details are mentioned, therefore it is not clear what is their biological importance. This is true for Fig1 as well . It is not clear what properties exactly are implied or how wtKRAS affects mKRAS biology etc etc

Answer: In the revised manuscript, we describe more in detail that KRAS mutational isoforms differ in their biochemical properties to hydrolyze GTP and to activate downstream signaling pathways. This determines their biological behavior and affects therapeutic vulnerabilities as e.g. sensitivity to MEK inhibition. Furthermore, we describe more in detail the effect of wildtype KRAS on mutant KRAS through dimerization. This dimerization affects the transforming potential of mutant KRAS and impacts therapeutic interventions (e.g. MEK inhibition).  

Fig 2 and 3 the description is very vague, references in the text not well marked either. As a result both figures are not as helpful for the reader.

Answer: We have added the respective references in brackets to Figures 2 and 3 to make it more clear for the reader where to find the description of KRAS-mediated effects on the tumor immune microenvironment. We also specified the description of Figures 2 and 3 more.

Round 2

Reviewer 1 Report

No further comments